# Self-Healing Mechanical Properties of Selected Roofing Felts

**DOI:** 10.3390/ma16031204

**Published:** 2023-01-31

**Authors:** Bartosz Łuczak, Wojciech Sumelka, Wojciech Szymkuć, Hubert Jopek

**Affiliations:** 1Institute of Structural Analysis, Poznan University of Technology, Piotrowo 5, 60-965 Poznań, Poland; 2Institute of Applied Mechanics, Poznan University of Technology, Jana Pawła II 24, 60-965 Poznań, Poland

**Keywords:** self-healing, roofing felt, mechanical properties, tension test

## Abstract

In this work, roof felts are considered. Special attention is paid to the mechanical properties and self-healing (SH) phenomena under elevated temperatures. The results of the heating and strength tests for the entire range of material work, from the first load to sample breaking, are shown with respect to the angle of reinforcement relative to the longitudinal axis of the sample and different ways of breaking the continuity of the material. The influence that the material thickness and modifiers used for the production of the base material have on the obtained results was also pointed out. The meaningful SH strength is reported—from 5% up to 20% of the strength of the undamaged material—which, in perspective, can provide comprehensive knowledge of the optimal use of roofing felts and its proper mathematical modeling.

## 1. Introduction

In recent years, self-healing (SH) in different types of materials has gained popularity and is undergoing extensive research. Materials with SH properties may be a revolution in almost any industry, i.e., in electronics, allowing, for example, for the invention of new, better sensors or batteries with SH housing, in medicine, or even in civil engineering. The research presented in this work focuses on determining the mechanical properties of the SH of different types of roofing felt under certain conditions, and on materials without any modification. Roofing felts are still one of the main materials used as waterproof insulation in commercial and residential buildings. Therefore, a better understanding of the material is essential for further development and new design methods.

The papers reviewed may be divided into a few groups: those presenting new modifications to different types of materials, those describing new material models, and those describing different approaches to the mathematical modeling of materials with SH properties.

In the group of articles that present the results of the experimental research of new materials, one can distinguish the work by [1]. The authors present the results of the 3D printing material with SH properties [2]. Materials with these properties are also presented in [3]. In addition, some materials mimic the transparent, luminescent, and SH properties of nature [4]. Next, in [5,6,7,8], the new materials that may find an application in wearables, stretchable sensors in the field of healthcare, and electronic skins are investigated. Another field where materials with SH abilities may be valuable is medicine [9], where the SH ability of the presented nontoxic hydrogel, reinforced with montmorillonite, makes it a valuable medical treatment. Recently, the abilities of SH gained attention in energy storage applications, and general information can be found in [10].

Furthermore, the SH properties of building materials are especially important in terms of their durability [11]. In [12,13], a review of SH concretes (SHC) is presented, and in [2], a review of crystalline admixtures and their impact on the SH of concrete is presented. An improvement in SH abilities and a simultaneous reduction in shrinkage using microcapsules is presented in [14]. Concrete and mortars with SH properties, with the possibility of the cracks’ closure, are presented in [15,16,17,18,19,20,21,22]. The methodology for evaluating concrete’s self-healing performance is presented in [23]. In [24], the SH properties of special concrete bricks are presented, and in [25], the SH capacity and behavior of fiber-reinforced concrete materials are discussed under static and cyclic loading. There are also some articles that present an extensive overview of SH materials, i.e., [26], for self-healing concrete [27,28,29]. Furthermore, the reinforcement of concrete can have SH properties; the characterization of this type of material is presented in [30]. In [31], the approach to controlling steel reinforcement corrosion with SH is discussed. Another application may be different types of SH coatings [32]. In civil engineering, such materials could solve the problem of metal corrosion. Some strategies and methods for corrosion protection are reviewed in [33]. Two different approaches can be distinguished for metals with SH properties [34] and coatings with such properties [32,35,36,37,38]. Special polymer coatings may be used to secure other materials [38]. From the perspective of roofing felt, due to the main material from which the roofing felt is made, very promising research is presented by [39], where the authors present the result of the properties of SH in recycled asphalt. The authors of [40] present SH in microcapsule asphalt; [41] presents SH in asphalt due to the microwave heating of the pavement. The durability and aging of self-healed asphalt concrete is discussed in [42], and the performance of SH in porous asphalt is presented in [43]. Modeling SH materials is another group of articles that could be distinguished. SH material models are presented in [44,45] and those for polymeric materials in [46,47]. A novel constitutive elastoplastic model of SH materials is proposed in [48]. The thermodynamic model of the damage-healing process is presented in [49]. Another constitutive model of SH materials is presented in [45], anisotropic models are presented in [50,51], and a wound model is presented in [52].

The numerical modeling of materials with SH could be divided into two groups of simulations, one for modeling the behavior of the material and structures made of the material with SH and the other focusing on the numerical design of the material with SH, which will fulfill certain properties. The first group could be represented by [53,54], in which the authors present the self-healing behavior in cement-based and concrete material. In the former context, interesting research is presented in [55,56], in which the authors try to predict the life-cycle of self-healed products. In [57], a numerical simulation of the response of concrete structural elements containing an SH agent is presented. In [58], the use of a neutral network is proposed to model the SH process. A simulation of the SH performance of autogenous concrete using machine learning is presented in [59]. In [60], a simulated-aided design of the capsules for SH concrete is presented, whereas, in [61], the simulations and experimental investigations of the mechanical properties and solubility of 3D-printed capsules for SH cement composites are presented.

The conducted literature review indicates researchers’ great interest in topics related to materials with SH. The object of this interest is a large group of building materials; however, the lack of interest and valuable research carried out in the field of SH on one of the basic insulating materials, which is roofing felt, is clearly visible. The main objective of this work is to investigate the SH properties of roofing felt.

This article is a continuation of the work started in [62]. However, in this article, an attempt is made to assess the possibility of connecting damaged, not modified in any way, roofing felt and to evaluate the strength of the joint resulting from material self-healing as a result of exposing the damaged material to an elevated temperature. In the next section, more details about the tested materials are given. In Section 3, details about the methodology of preparing the test samples and testing are presented, and in Section 3.2, information about the heating procedure is presented. Section 4 contains the results and a discussion, while the last chapter summarizes the work.

## 2. Analyzed Materials

The purpose of this research is to determine the residual tensile strength of roofing felt samples that were subjected to a tensile test in the entire range of its work, namely from elastoplastic up to tearing of the sample. Moreover, for comparison, samples cut with a sharp knife are also considered. Four different types of material are examined, varying in thickness, base material, and surface finishing methods. The results are presented for materials that were not modified to obtain any type of SH property. The total secondary strength is the result of the sum of the basic properties of the constituent materials. The signatures of these materials are introduced as follows: M1, M2, M3, and M4. For three of the four types of tested material, the matrix was made of the same material, and this material was nonwoven polyester (materials M1, M2, and M3). The matrix was made of a glass veil in the fourth material (M4). Two out of four types were made of elastomer-modified bitumens (M1, M3), one was made of synthetic modified bitumen rubber (M2), and one was made from oxidized bitumen (M4). The thickness of the analyzed products are 2.2 mm (M2, M4) and 5.2 mm (M1, M3). Material details are presented in Table 1. The thickness given for all samples was within the tolerance declared by the manufacturer for materials M1 and M3, ±0.5 mm, and for materials M2 and M4, it was ±0.4 mm. Detailed information on the selected materials, their chemical composition, microscopic structure, and mechanical properties are presented in previous work [62].

## 3. Methodology

### 3.1. Samples Preparation

This research focused on exploring the possibility of creating a force-transmitting connection by heating two series of samples. In the first series, samples were prepared for tensile strength tests according to the method and dimensions presented in [62]. As a result, specimens of three types were cut from a sheet of roofing material at an angle of 0°, 45°, 90° to the reinforcement direction of the material for each one of four materials (M1–M4) obtained. This allowed to study the influence of the reinforcement matrix on the results. Then, half of them were cut into half with a sharp knife. Thus, a straight, even joint surface in the narrower part between the two halves was created. The samples before and after division are presented in Figure 1.

The second series of samples were prepared in the same way as previously described, except for cutting, and were subjected to a tensile test until tearing was caused according to the procedure presented in [62]. Therefore, a random and nonuniform connection area was created between the sample halves. Due to the heterogeneous nature of these surfaces, as determined by different angles and shapes, it was impossible to reliably evaluate the size of the area that participates in the connection. Therefore, this value is not specified in the paper. An example of samples obtained in this way is shown in Figure 2.

Then, all the samples were heated, as described in detail in Section 3.2. Afterward, the samples that had successfully merged were subjected to a second tensile test. Only samples made of materials M1 and M3 were connected in each analyzed case. Specimens made of the M2 material as a result of heating were only connected for samples previously subjected to a tensile test. It was not possible to combine samples made of M4 material in any of the analyzed cases; therefore, the results for this material are not described further in this paper.

The strength test was carried out with a constant displacement speed equal to 5 mm/min, until the sample broke or there was a sudden drop in force value of about 80%. During the experiments, the force and displacements of the tensile grips were recorded. The experiment was repeated at least five times for each type of sample from each material, at an angle of 0°, 45°, 90° to the reinforcing matrix and for samples cut with a knife after the tensile test.

### 3.2. Heating Process

To simulate the SH process, the specimens underwent a heating process. The roof material goes through a similar process every day. On summer days, the surface temperature of the roof membrane may reach about 60 °C [63]. Under such conditions, most roofing membranes change properties, become soft and plastic, and return to their previous state when the roof temperature drops.

A forced convection climate chamber with two fans that forced air to circulate was used to simulate heating of the roofing felt samples. The decision to use this device was motivated by the possibility of a more precise temperature control in the tested range below 100 °C, where the temperature is also more stable, and even throughout the chamber due the forced flow of air. The chamber during the tests is shown in Figure 3.

The heating process was divided into parts so that there is always only one layer of samples in the chamber. To prevent the samples from sticking to the racks of the device and reduce the amount of air directly flowing from the bottom of the samples, they were placed on sheets of non-stick paper adapted to work at elevated temperatures. The samples were placed in such a way that both halves touched each other, forming a secondary connection between them. The arrangement of the samples before the start of the heating process for one of the series is shown in Figure 4.

During the experiment, the temperature was measured at three different points: below the rack with samples, above the rack with samples, and on the surface of the samples. Temperature was measured with K-type thermocouples (class 1 according to EN 60584-1) and recorded with a device with cold junction temperature compensation. The procedure consisted of three stages to roughly simulate the course of temperature changes on the roof surface during the summer day. In the first stage, the samples were heated from room temperature (25 °C) until they reached 75 °C. Then, the temperature of 75 °C was kept constant for 1.5 h. Finally, the heating was turned off, the chamber doors were opened, and the samples were allowed to cool to room temperature (approximately 25 °C). The changes in temperature in the chamber during the experiment are shown in Figure 5. The four apparent sudden temperature drops are the result of opening the chamber door to evaluate the condition of the samples during the experiment.

After heating and cooling down, the samples were conditioned at room temperature in dark conditions for another 48 h. The heating procedure had no effect on the dimensions of the samples, while the measurements before and after the process provided dimensions within the measurement tolerance ±0.5 mm. The samples after the heating process are presented in Figure 6.

## 4. Results

The first aspect worth noting is the behavior of the material during the attempts to create a connection under the heat treatment and the way the contact surface area was created. In the first series of tests, in which the samples were cut in half with a sharp knife, creating a homogeneous, smooth, and even cutting surface, the connection was made only in two of the four materials, materials with a thickness greater than 5.2 mm (materials M1 and M3). This indicates a clear influence of the thickness of the material on the possibility of obtaining a joint—a thicker material creates stronger joints—which should be further investigated on a larger number of materials of different thicknesses. In the second series of studies, when the joint was caused by the tearing of the samples during the tensile tests, the joint surface was random, uneven, and nonuniform; however, in this case, it was possible to achieve bonding by the temperature for three of the four materials (materials M1, M2, and M3).

In the cases where a connection was established, an attempt was made to evaluate the strength of the bond. The samples were carefully placed in a test machine and then subjected to a tensile test with a constant displacement speed equal to 5 mm/min, until the sample broke or there was a sudden drop in force, during which the force and displacement were measured. All the tested samples that made a connection as a result of the heat treatment showed a strength equal to approximately 5% up to 20% of the primary samples, which was not damaged in any way.

Each figure shows two graphs. The blue one shows the results for the primary samples, i.e., those not subjected to a tensile test, cut, or heat treated, with the corresponding legend on the right and top. The second graph, in green, and the corresponding legends on the left and bottom show the results of the tensile test for the samples that were cut and annealed in Figure 7 and Figure 8. Figure 9 and Figure 10 present the results of the samples that were first subjected to a tensile test and then heat treated. On this basis, it can be concluded that, for the samples cut with the knife, M3 provides a higher maximum force compared to the material M1. This is the opposite of the results for the primary samples, where the material M1 was characterized by a higher maximum force value. The difference in behavior is probably due to the difference in the chemical composition of the roofing felt base material and the modifiers used by the manufacturer, which is also worth checking in future works. No distinct trend was observed for the displacements. Similarly, for samples after the tensile test and heated, the same relationships at the maximum force that were obtained for the material in primary tests cannot be found, e.g., for samples cut at an angle of 45° to the angle of the reinforcement material, M2 provides the maximum value of the breaking force in the second test, but in the primary ones, the M2 material was the weakest compared to M1 and M3.

However, the behavior of the sample cut with a knife for a single material with respect to the angle of reinforcement relative to the longitudinal axis at 0°, 45°, and 90° provides similar results to those obtained during the primary sample tensile tests. The lowest strength was observed for the 45° angel, median values were obtained for 0° and the highest values for 90°. This indicates that, even after cutting the mesh of the reinforcing fibers, they still affect the behavior of the rest of the sample, probably helping to spread the forces during the experiment. Nevertheless, there was no effect on the results obtained for different types of materials from which the reinforcing matrices were made. However, a similar relationship cannot be stated for samples after the tensile test, for which the results are random.

The data in the graphs also clearly show that the samples that were cut with the knife and then joined have a higher strength compared to the samples after the tensile tests. In some cases, the samples obtained in this way were twice as strong compared to the samples obtained from the tension test. This is a probable effect of the different types of contact surface. Another conclusion could be that because the reinforcement matrix in both cases was discontinued, this may indicate that, in the test, the strength of the base material itself was determined without the influence of the reinforcement layer—the strength of healthy, undamaged material is described in detail in the previous work [62]. The lack of a reinforcement matrix also results in a less significant difference in strength, depending on the direction in which the samples were cut—the effect of the reinforcement matrix on the anisotropy of the material is described in [62], which indicates the possibility of examining the impact of modifiers used in the production of the base material of roofing felt. In both cases, there are big differences in the strength for each material. This may be due to the heterogeneity of the material, which was shown in microscopic photos in [62], and the unconstrained method of the connection formation. There is also a noteworthy tendency that, despite their much lower strength, the samples obtained from the tensile test are generally characterized by a greater overall deformation. The samples cut with a knife deteriorate more rapidly, without significant elongation.

## 5. Conclusions

The presented research methodology and results focus on determining the mechanical properties of the self-healing (SH) of different types of roofing felt under specific conditions. Roofing felt is still one of the main materials used as waterproof insulation in commercial and residential buildings. Therefore, a better understanding of the material is essential for the further development of the material and new design methods.

This publication extends the methodology and testing procedure presented in the previous article. A detailed description of the preparation of the samples and the way of conducting the experiments is presented. In this work, the main interest is the assessment of the possibility of creating a connection between the two halves of the samples under certain conditions and evaluating the strength of such a connection using the methodology proposed earlier. As presented, the establishment of such a connection is possible; therefore, the SH of roofing felts is possible as a result of exposing the samples to temperature. Furthermore, such a connection may be characterized by a quite significant strength. The experiments show that the samples that were cut with the knife and then joined have a higher strength compared to the samples after the tensile tests. Furthermore, the obtained strength depends on the contact surface. For the samples with even and straight contact areas, the maximum breaking force in some cases may be two times higher than in an analogous sample after the tensile test. However, despite the much lower strength of the samples obtained from the tensile test, they are generally characterized by a greater overall deformation. The samples cut with a knife deteriorate more rapidly, without significant elongation. The type of contact surface also significantly influenced the obtained results. The samples after the tensile test showed a much greater difference in the results, not allowing for the patterns present in the previous primary tests to be distinguished. The samples cut with a knife provided less random results, and it was possible to distinguish similar patterns of material behavior in the case of primary tests. As the reinforcement matrix in both cases was cut, this may indicate that, in the test, the strength of the base material itself was determined without the influence of the reinforcement layer. Hence, the randomness of the results may be due to the heterogeneity of the material and the unconstrained method of connection formation.

One can conclude that the mechanical anisotropy in the samples obtained as a result of sample discontinuity and its reconnection are less observable compared to the results in which the sample was not broken. It should be pointed out that the results are fundamental from a theoretical point of view, in the sense of the strategy used for the computational modeling of SH processes. This will be the next step in the research. In future works, we would also like to analyze the impact of additives and modifiers used in the production of the base material of roofing felt, as well as check other ways of heating samples and how that impacts the strength of the connection.

## Figures and Tables

**Figure 1 materials-16-01204-f001:**
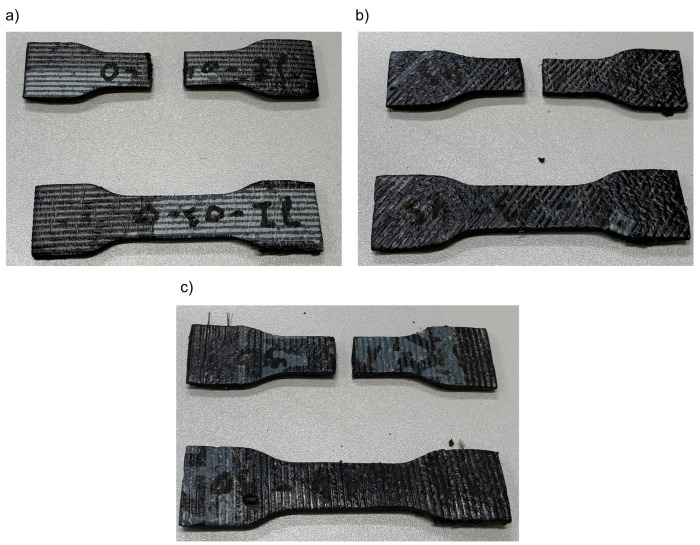
M1 material samples prepared according to [62] with (**a**) 0°, (**b**) 45°, (**c**) 90° angle of reinforcement relative to the sample’s longitudinal axis. Samples cut with a knife.

**Figure 2 materials-16-01204-f002:**
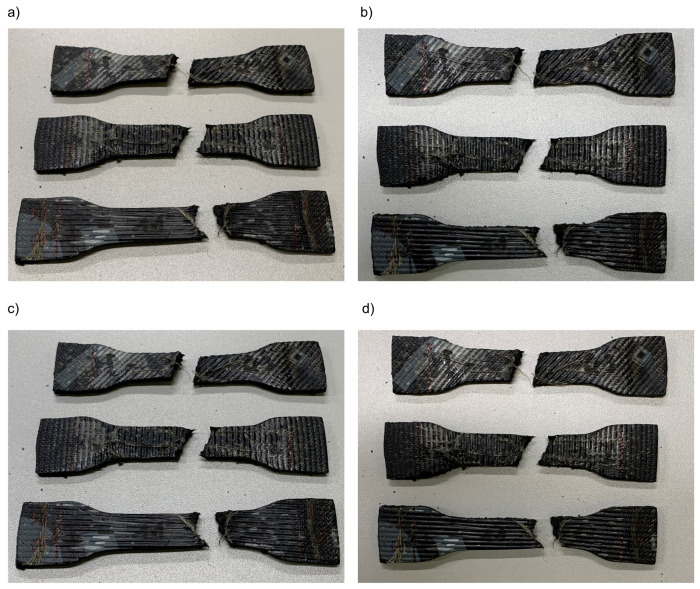
M1 material samples (**a**–**d**) prepared according to [62] with 0°, 45°, 90° angle of reinforcement relative to the sample’s longitudinal axis. Samples after the tensile test.

**Figure 3 materials-16-01204-f003:**
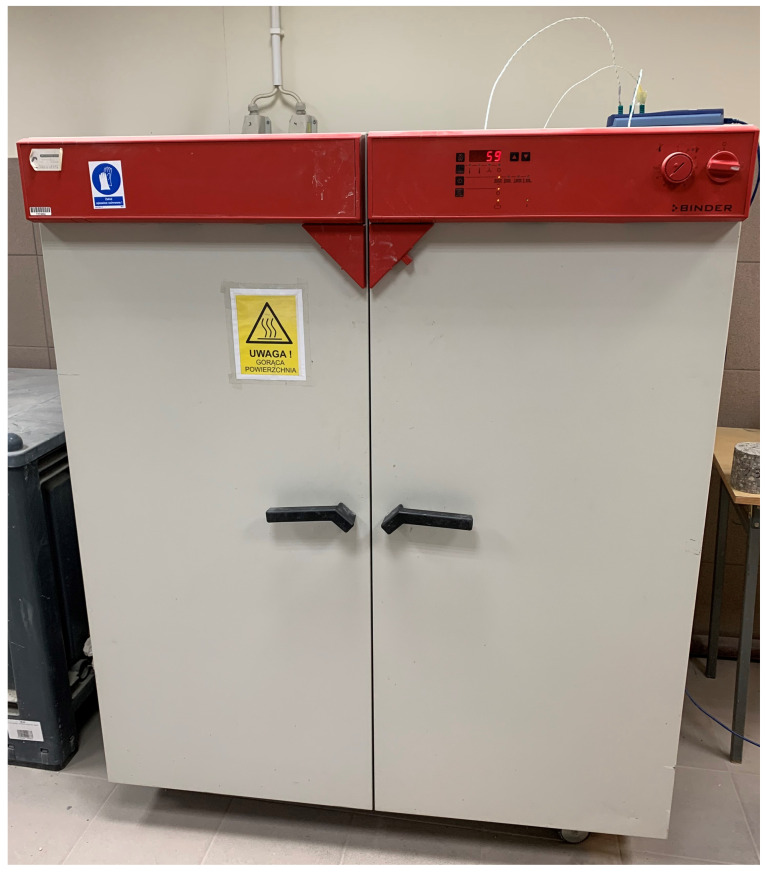
Climate chamber during the heating procedure of samples.

**Figure 4 materials-16-01204-f004:**
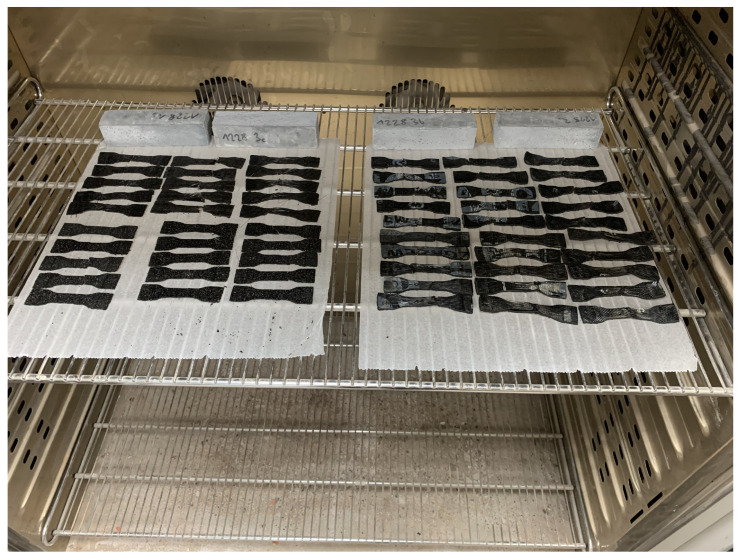
Samples in the climate chamber, arrangement before the heating.

**Figure 5 materials-16-01204-f005:**
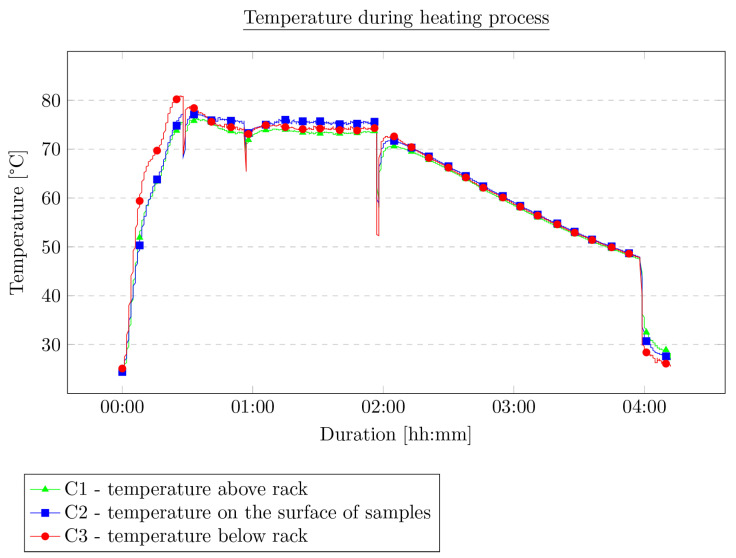
Temperature course during the experiment.

**Figure 6 materials-16-01204-f006:**
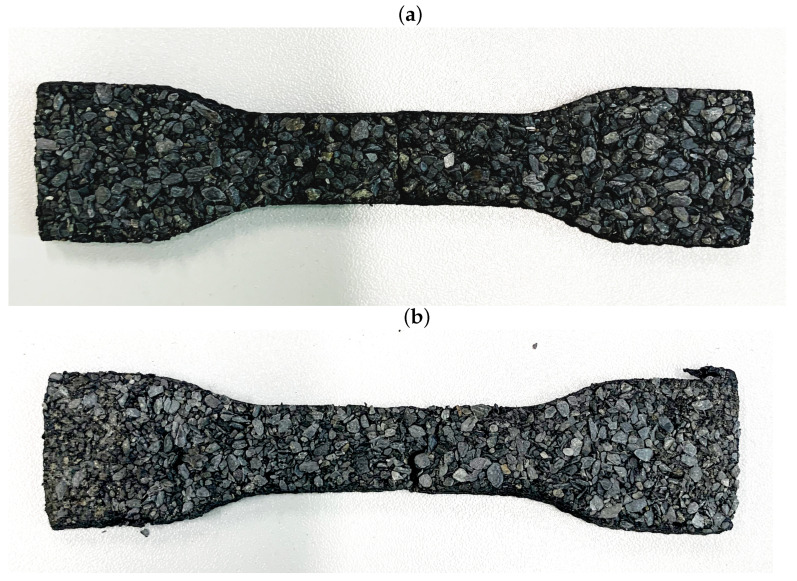
Samples after SH process: (**a**) sample cut a with knife, (**b**) sample after tensile test.

**Figure 7 materials-16-01204-f007:**
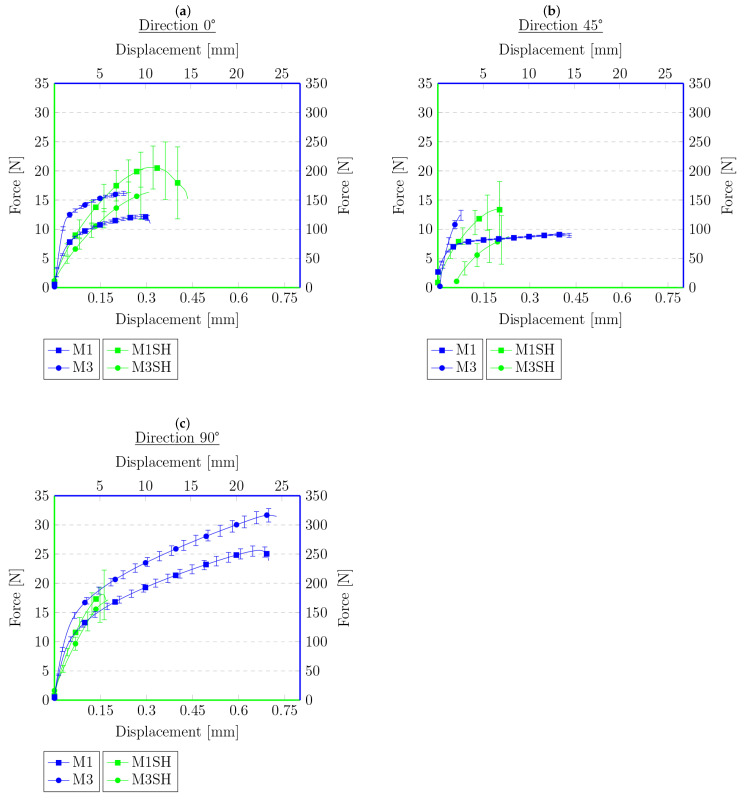
Force–displacement graphs for primary samples (blue) and samples cut with a knife and heated (green) with respect to the angle of reinforcement relative to the sample’s longitudinal axis: (**a**) 0°, (**b**) 45°, (**c**) 90°.

**Figure 8 materials-16-01204-f008:**
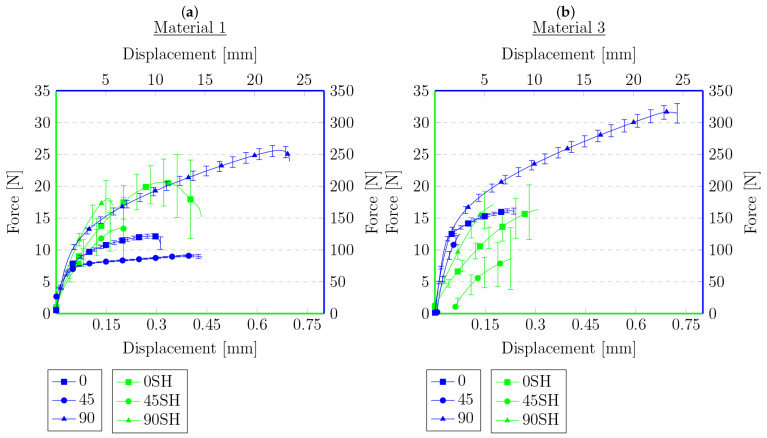
Force–displacement graphs for primary samples (blue) and samples cut with a knife and heated (green) for materials for which connection was established with respect to the angle of reinforcement relative to the sample’s longitudinal axis: (**a**) 0°, (**b**) 45°.

**Figure 9 materials-16-01204-f009:**
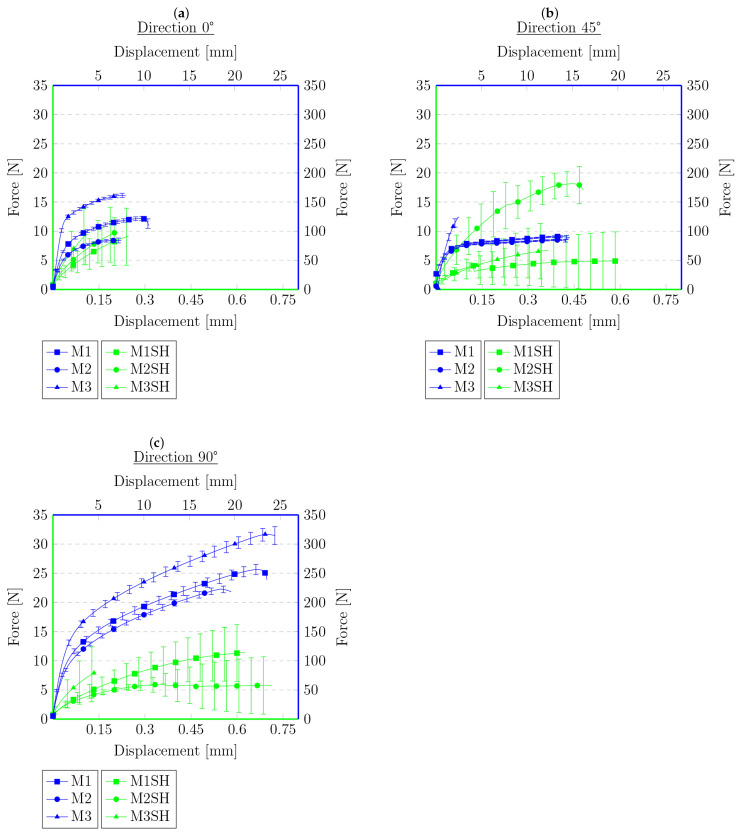
Force–displacement graphs for primary samples (blue) and samples after tensile test and heated (green) with respect to the angle of reinforcement relative to the sample’s longitudinal axis: (**a**) 0°, (**b**) 45°, (**c**) 90°.

**Figure 10 materials-16-01204-f010:**
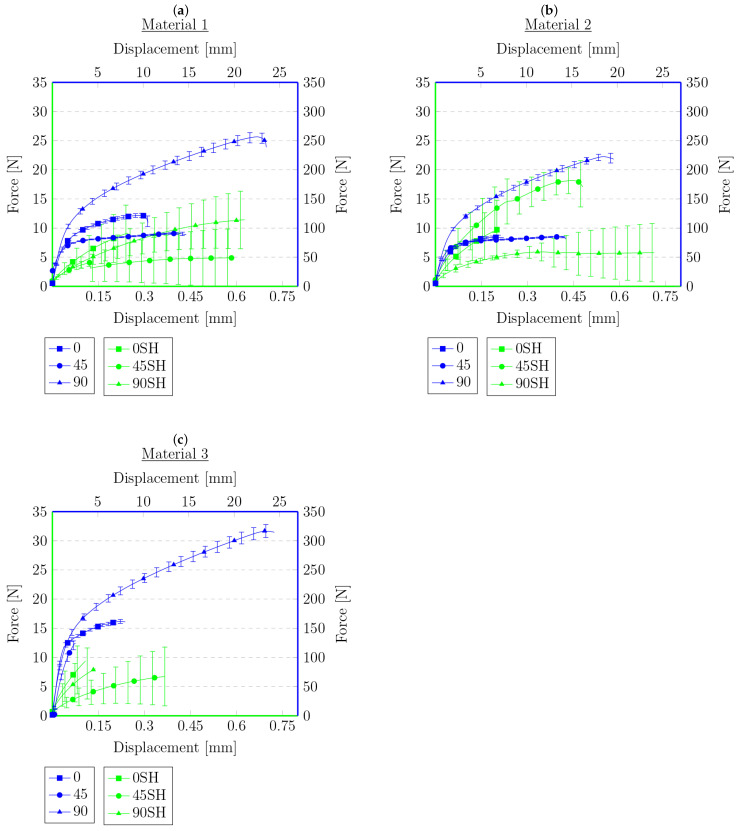
Force–displacement graphs for primary samples (blue) and samples after tensile test and heated (green) for materials for which connection was established with respect to the angle of reinforcement relative to the sample’s longitudinal axis: (**a**) 0°, (**b**) 45°, (**c**) 90°.

**Table 1 materials-16-01204-t001:** Analyzed materials properties.

Property Name	M1	M2	M3	M4
Matrix material	nonwoven polyester			glass veil
Base material	elastomer-modified bitumen	synthetic modified bitumen rubber	elastomer-modified bitumen	oxidized bitumen
Thicknes [mm]	5.2 ± 0.5	2.2 ± 0.5	5.2 ± 0.5	2.2 ± 0.5

## Data Availability

The data presented in this study are available on request from the corresponding author.

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
