# Peer review of "Self-Healing Mechanical Properties of Selected Roofing Felts"

_materials, 2023, doi:10.3390/ma16031204_

Round 1

Reviewer 1 Report

All materials are vulnerable to natural or artificial degradation and deteriorate with time. In the case of structural materials, the long-time degradation process leads to microcracks that cause failure. Thus, repairing is vital to enrich the reliability and lifetime of materials. The present study enriches the literature in this domain. The reviewer has observations that need to incorporate in the revised manuscript for the enhancement of quality.

What type of self-healing mechanism is used in the present investigation? Explain.

The author has conducted an extensive survey of the literature. But the research shortfall and objectives of the present research were missing.

How the thickness of the material for the analysis purpose was selected as 2.2 mm and 5.2 mm?

Could you give a schematic representation of sample preparation in detail?

Why reinforcement are aligned at 0°, 45°, and 90° relative to the sample’s longitudinal axis? Is there any physics?

‘The lowest strength was observed for the 45° angel…… the highest values for 90° Why?

What are the reasons for mechanical anisotropy in the present investigation? How it can be minimized?

What materials are available as roofing felt? How does this material outperform other choices?  

Author Response

  1. At this stage of the investigation, we concentrate on the phenomenological aspect of SH. Detailed studies on certain physical mechanisms are in future plans.
  2. The originality of the presented manuscript is summarised in the last paragraph of Section 1: “This article is a continuation of the work started in [ 62 ] (https://www.mdpi.com/1996-1944/14/22/6907 ). However, in this article an attempt is made to assess the possibility of connecting damaged, not modified in any way, roofing felt and to evaluate the strength of the joint resulting from material self-healing as a result of exposing the damaged material to elevated temperature.”
  3. The thickness of the roofing felts is generally between 2 to 5 mm, and in this range, 2.2mm and 5.2mm are the most common thickens across different manufacturers in Poland.
  4. The process of preparing samples is described in detail in our previous work entitled “Experimental Analysis of Mechanical Anisotropy of Selected Roofing Felts” [62], and therefore we have decided not to repeat these data.
  5. These are the two main directions for reinforcing matrix and the third - 45 degree is the value between these two perpendicular directions. This was done to check the behaviour of the material for forces acting not in the main longitudinal direction of the roofing felt - although in engineering practice direction 0 and 90 are most important.
  6. and 7. There are two main factors for anisotropy in this material. The first of course is the reinforcing matrix which has different strengths in both directions and the second non-uniformity of the material which was discussed in more detail in the previously mentioned work [62].
  7. Our aim was to study the roofing felt that is traditional, cheapest and thus the most used roofing material. However, there are more modern but also more expensive alternatives such as PDM, PVC, TPO membranes - this aspect could be a topic of future research.

Reviewer 2 Report

Dear authors,

I have reviewed the manuscript entitled Self-healing mechanical properties of selected roofing felts. In this article, the experiment presented the mechanical properties of self-healing (SH) of different types of roofing felts under conditions, that roofing felts were cut by a knife or processed by another method in which a tensile strength test was conducted and then heated, with four type materials and three different angles of reinforcement relative to the sample’s longitudinal axis.

The article is of interest both on a scientific level, and especially for the engineering industry to the SH properties of building materials. It is also useful for the assessment and examination of waterproof insulation of roofing felts. The paper needs some improvements on the following issues.

In general:

1. The structure of the article needs to be adjusted. For example, Figure 7-10 should be better placed in Results to facilitate the understanding of the analysis of the results

2. In page 2 line 61-72, there are two groups of simulations of numerical modeling of materials with SH written in the article, but only the former was mentioned. Please conclude the other group, just like the former does.

3. More details of the experiment overview should be introduced. As we all know, as for the discreteness of the experiment and the materials, the experiment results will be varied from each other to certain extents. How many times have the experiments in the article been repeated and whether the experiment results could be repeated?

4. The influencing factors of the experiment need to be comprehensively summarized and specifically presented. Especially important, is it the single action of Thickness, or the combined effect of base material and matrix material, even of three factors? And how do three factors affect the experiment results independently and jointly?

Details:

1. In page 3 line 96, there is a print mistake –" Table ??."

2. In page 3 line 104, the four materials (M1-M4) were obtained from a sheet of roofing material. But in page 3 Table 1, the properties of the four materials are different. It seems contradictory Please explain how the different four materials (M1-M4), according to thickness, base material, and matrix material, are obtained.

3. In page 3 Table 1, the four types of material samples were designed, but M2 was bearly not involved, and even the force-displacement graph or discussion of M4 was completely ignored. Please complete the part about M4.

4. According to page 8 line 179-183, there are significant differences in force between the M1 and M2 among all kind of examinations including primary and secondary. But in page 3 Table 1, the properties of M1 and M2 are the same. Please explain it, and an elaborate experiment schedule should be given.

5. In page 8 line 217, " This publication extends the methodology and testing procedure presented in the previous article.", please brief the methodology and testing procedure. the extension in this article basing the previous articles should be illustrated, and the relationship with the previous articles should be specified.

Grammar problem:

1. Page 2 line 84: " elastoplastic up to tearing the sample " needs to be replaced with " elastoplastic up to tearing of the sample ".

2. Page 2 line 92: " analyzed " needs to be deleted.

3. Page 3 line 95: " was " needs to be replaced with "is ".

4. Page 3 line 95: " was " needs to be replaced with ", and were ".

5. Page 3 line 96: " present " needs to be replaced with " presented ".

6. Page 8 line 178: " are " needs to be deleted.

7. Page 8 line 184: " between " needs to be replaced with "at ".

8. Page 8 line 188: " comparing " needs to be deleted.

9. Page 8 line 214: " The roofing felts are still one of the main materials " needs to be replaced with " The roofing felt is still one of the main materials ".

10. Page 8 line 218: " was " needs to be replaced with "is ".

11. Page 10 line 221: " a result of exposure of samples to temperature " needs to be replaced with " a result of exposing samples to temperature ".

12. Page 10 line 222: " shows " needs to be replaced with "show ".

13. Page 10 line 239: " is " needs to be replaced with " are ".

Author Response

General remarks:

  1. The structure will be adjusted during the final editorial process - thank you for this remark.
  2. Has been addressed.
  3. All details about the samples and experiment can be found in  previous work [62] (https://www.mdpi.com/1996-1944/14/22/6907 ); hence only new elements were added to this article to avoid unnecessary repetition.
  4. The influencing factors of the experiment has beed added in Sections 3 and 4.

Details:

Thank you very much for such detailed remarks. All comments has been addressed.

Grammar problem:

The article has been subject to language verification by MDPI group.

Reviewer 3 Report

The manuscript is defiantly shown a great effort of experiments and it is worth to be published. However, I would like to address the following questions or comments to be taking in consideration when revising the manuscript:

 1. It is better to consider expanding your abstract with adding experimental results.

 2. Lines 96: Which table does “ Table ?? ” refer to? Please number the table correctly.

 3. Lines 98: Please cite the literature correctly. Please rephrase the sentence “ More information on the selected……….”.

 4. Lines 146: What is the reference for heating the sample to 75 ℃ ? Please add a citation.

 5. It is better to consider improving the quality of images in the paper.

 6. It is recommended to use the professional drawing software Origin for drawing. The figures are uniformly chosen in Roman font.

 7. It is better to have this paper extensively edited to optimise the language.

 8. The introduction part is not fully cited, and a lot of fiber research has been carried out, for example: (1) Huan Zhang, Shuai Cao, Erol Yilmaz. Influence of 3D-printed polymer structures on dynamic splitting and crack propagation behavior of cementitious tailings backfill [J], Construction and Building Materials, 343(2022) .

 9. The author should consider expanding your conclusion with some directions for future work.

Author Response

  1. Some additional sentences are added in abstract to clarify the point, namely “The influence of material thickness and modifiers used for the production of the base material on the obtained results was also pointed out. The meaningful SH strength is reported - from 5\% up to 20\% of the strength of the undamaged material, which, in perspective, can provide comprehensive knowledge of the optimal use of roofing felts and its proper mathematical modeling.”
  2. Has been addressed.
  3. Has been addressed.
  4. Has been addressed.
  5. Has been addressed, and moreover we will take a special care during production with editorial team.
  6. Has been addressed.
  7. The article has been subject to language verification by MDPI group.
  8. Has been addressed.
  9. Has been addressed.

Reviewer 4 Report

The presented manuscript reports on experimental results documenting the mechanical properties and self-healing performance of roofing felts.

Please consider the following recommendations, hoping they might be helpful in improving the paper:

1) The abstract should contain quantitative findings. As it is, the abstract is too general.

2) The introduction should be re-written. As it is, it doesn't introduce the topic to the reader. It doesn't explain the mechanisms of self-healing or what actually happens at the material level but only lists applications without a precise reference to the quantitative performance.

A reader not familiar with the topic would have to go and read the articles you cited, which reduces the utility of the introduction.

3) A characterization of the analyzed materials - beyond a qualitative description and thickness - would be required. At least higher-quality images with a ruler should be supplied.

4) Heating process. The temperature distribution across the sample might be uneven, with higher temperatures at the exterior surface of the roofing felt, than at the surface in contact with the roofing deck. Heating produced by solar radiation might result in different effects. The representativity of the heating procedure should be considered and discussed.

5) SEM images would be beneficial.

6) Additional experiments to fully document the dynamics would be required.

Author Response

  1. Some additional sentences are added in abstract to clarify the point, namely “The influence of material thickness and modifiers used for the production of the base material on the obtained results was also pointed out. The meaningful SH strength is reported - from 5\% up to 20\% of the strength of the undamaged material, which, in perspective, can provide comprehensive knowledge of the optimal use of roofing felts and its proper mathematical modeling.”
  2. At this stage of the investigation, we concentrate on the phenomenological aspect of SH. Detailed studies on certain physical mechanisms are in future plans. Moreover, all details about the samples and experiment can be found in  previous work [62] (https://www.mdpi.com/1996-1944/14/22/6907 ); hence only new elements were added to this article to avoid unnecessary repetition.
  3. As mentioned in poit 2, ll details about the samples and experiment can be found in  previous work [62] (https://www.mdpi.com/1996-1944/14/22/6907 ).
  4. Thank you for this comment. Of course, our tests are in experimental conditions, and therefore the small thickness of the sample and surrounding conditions allow us to claim that in steady conditions the temperature was equal at each point. We will consider your comment in future experimental work as it is a hard task to build such an experimental set-up to be close to real roof conditions.
  5. As mentioned in poit 2, ll details about the samples and experiment can be found in  previous work [62] (https://www.mdpi.com/1996-1944/14/22/6907 ).
  6. We believe that at this stage our results are prepared in a very detailed manner, and moreover, the description allows for repetition by other groups. Nonetheless, the consideration of full analysis in time seems to be a very interesting point for further study. Thank you.

Round 2

Reviewer 2 Report

Dear Author

Now it could be accepted for publication. 

Author Response

We would like to thank You for taking the necessary time and effort to review the manuscript. We sincerely appreciate all your valuable comments and suggestions, which helped us in improving the quality of the manuscript.

Reviewer 3 Report

Accepted

Author Response

(The authors gave the same response as above.)

Reviewer 4 Report

This revised submission clarified some points and improved the manuscript, although

a) the introduction is still very qualitative

b) the discussion could be further improved by comparing your results and findings with those in the literature.

Author Response

  1. a) the introduction is still very qualitative

We have decided to add information on type of tested materials, namely “Four different types of material are examined: two made of elastomer-modified bitumens (with a nonwoven polyester matrix), one made of synthetic modified bitumen rubber (with a nonwoven polyester matrix), and one made of oxidized bitumen (with glass veil matrix).” 

  1. b) the discussion could be further improved by comparing your results and findings with those in the literature.

To the best of our knowledge, there is no available literature data for materials presented in this manuscript. Other materials (concrete, asphalt etc.) that are available are mentioned in the introduction part.  Nonetheless,  some extensions are added in the introduction to make the originality more pronounced.